# Sexual Rewards and Costs in Heterosexual and Gay Hispanic Adults

**DOI:** 10.3390/healthcare12020170

**Published:** 2024-01-11

**Authors:** Cristobal Calvillo, Juan Carlos Sierra, María del Mar Sánchez-Fuentes, Reina Granados

**Affiliations:** 1Department of Health Behavior & Health Education, Fay W. Boozman College of Public Health, University of Arkansas for Medical Sciences, Little Rock, AR 72205, USA; cfcalvillomartinez@uams.edu; 2Mind, Brain, and Behavior Research Center, University of Granada, 18011 Granada, Spain; jcsierra@ugr.es; 3Nursing Department, Faculty of Health Sciences, University of Granada, 18071 Granada, Spain; reina@ugr.es

**Keywords:** sexual satisfaction, sexual costs, sexual rewards, Hispanics

## Abstract

The Interpersonal Exchange Model of Sexual Satisfaction proposes that sexual satisfaction can be explained by the balance of sexual rewards or costs exchange. The Interpersonal Exchange Model of Sexual Satisfaction Questionnaire (IEMSSQ) was developed from this model. The IEMSSQ includes the Reward/Costs Checklist (RCC). The RCC assesses different sexual exchanges that are categorized into sexual rewards and/or costs. Analyses and comparisons of sexual rewards and costs in the Hispanic cisgender population based on gender (man or woman) and sexual orientation (heterosexual and homosexual) with this measure have not been conducted. The main goal was to analyze and rank the sexual rewards, costs, and both, indicated by a sample of 1996 Hispanic partnered participants (heterosexual men, gay men, heterosexual women, and lesbians). The predominant sexual exchanges that were reported involved emotional aspects, both as rewards and costs. When participants indicated that the exchange was both a reward and a cost, they were referring to aspects specifically related to the sexual relationship itself. Heterosexual men and women had greater rewards and higher costs, respectively. Gay men obtained a greater number of items reported as rewards and costs. The results provide further insight into sexual satisfaction related to gender and sexual orientation.

## 1. Introduction

Social Exchange Theories are based on the economic theories of rewards and costs. In this way, just as an economic exchange is motivated by profit, social exchange is influenced by the rewards that these relationships can bring [1,2,3]. The purpose of these theories is to predict and explain human behavior, considering the aspects that are positive (rewards) and negative (costs). According to these theories, individuals prefer reward exchanges to cost exchanges. People carry out a rewards–cost analysis, wherein they evaluate the pros and cons of their interpersonal relationships [4].

In the sexuality field, these types of exchange theories can be very useful in explaining sexual relations because they consider the influence of the interpersonal context [5]. Despite their relevance in the sexuality context, very few researchers have used these theories for their studies. However, some have considered that Social Exchange Theories explain traditional gender roles and sexual behaviors [6,7]. In this sense, sexual relations in heterosexual people can be understood as a context in which men acquire sex from women by offering resources in exchange. Therefore, for these societies, traditional gender roles are defined as if men were the acquires or receivers and women were the providers or contributors to these interactions, thus giving value to factors such as virginity or fidelity in the case of women but not in men [6,7]. Social Exchange Theories have also been applied to the study of the selection of partners [8], the beginning of activity and sexual rejection [9], extradyadic sexual behavior [10], and sexual coercion [11]. In addition, some authors have considered using Social Exchange Theories to develop specific theoretical models related to sexuality, such as the Equity Theory [12], the Investment Model [13,14], and the Interpersonal Exchange Model of Sexual Satisfaction (IEMSS) [15,16].

The IEMSS, developed from Social Exchange Theories [3], is one of the most outstanding models for the study of sexual satisfaction due to its consistent empirical support and its applicability in men and women [17,18,19,20,21,22]. According to this model, sexual exchanges are situations, behaviors, and/or thoughts that occur during sexual relations and can be considered positive or negative [15]. Positive exchanges (rewards) are those that are pleasurable or gratifying, while negative sexual exchanges (costs) are those that cause discomfort, pain, shame, or anxiety or require physical or mental effort [16]. Thus, it is understood that sexual satisfaction is the result of the general evaluation that individuals make of the exchanges experienced as sexual rewards and costs and not of specific exchanges. However, the evaluation of specific sexual exchanges can be useful for designing treatments and studying exchanges that are considered positive or negative for people. For example, Lawrance and Byers [16] found that the most frequently reported sexual rewards in a Canadian heterosexual sample were related to emotional and relational exchanges, as well as physical and behavioral exchanges, while more than 50% of the sample considered having sexual relations as a sexual cost when they or their partner were not in the mood. Another study carried out in a heterosexual Chinese population found no statistically significant differences between men and women in sexual exchanges that were considered rewards, but there were differences in exchanges perceived as costs. Men reported more physical and emotional exchanges as sexual costs compared to women [23]. Other research conducted in a gay Canadian population showed that women rated exchanges associated with emotional and relational aspects as rewards more frequently than men, while exchanges with physical and behavioral aspects were rated as sexual costs for these women [24]. The first study conducted in a heterosexual Hispanic population [20] found that men identified physical exchanges as rewards more frequently than women, whereas women identified these exchanges as sexual costs. However, coinciding with the study by Cohen et al. [24], women more frequently indicated exchanges that referred to emotional aspects as rewards [20]. Finally, it is worth mentioning two recent studies carried out on the Hispanic gay population, in which both studies revealed that gay men reported sexual intercourse as a cost more frequently than lesbians [25,26].

Given this, it can be stated that there are diverse results regarding sexual exchanges considered as rewards and costs depending on gender (i.e., men and women) and sexual orientation (i.e., heterosexual and homosexual). According to previous research [24], gender and culture are variables that can influence the consideration of the most prominent sexual exchanges by individuals. Likewise, sexual minorities may experience different types of exchanges, such as sexual rewards and costs, compared to heterosexual individuals. This can be attributed mainly to cultural aspects that provide little social legitimacy to same-sex relationships [27], as well as the presence of more similar gender socialization experiences in same-sex couples than in the case of opposite-sex couples. Additionally, same-sex couples tend to rely more on idiosyncratic expectations and goals for their relationships [28]. There are no previous investigations that have analyzed the most frequent sexual exchanges considered as rewards and costs indicated by Hispanic men and women who have sexual relations with people of the same sex and different sex. The present research aims to find out the most frequent rewards and costs based on gender and the type of relationship that individuals maintain. The specific objectives are: (a) to examine the sexual exchanges most frequently considered as rewards, costs and rewards and costs in a sample constituted by men and women who maintain a heterosexual couple relationship; (b) to analyze the sexual exchanges most frequently considered as rewards, costs and rewards and costs in a sample constituted by men and women who maintain a relationship with a person of the same sex; and (c) to examine if there are statistically significant differences in sexual exchanges among these four groups: heterosexual men, heterosexual women, gay men, and lesbians.

## 2. Materials and Methods

### 2.1. Participants

For the present study, a non-probability convenience sampling was used, and a total of 1996 Hispanic participants were recruited (1042 cisgender men and 954 cisgender women). The sociodemographic characteristics of each subsample are displayed in Table 1. To participate in this study, individuals had to meet the following inclusion criteria at the evaluation time: (1) aged 18 years or older; (2) Spanish language as their mother tongue; (3) cisgender identity; (4) sexually active; and (5) involved in a steady relationship for at least a three-month duration with another cisgender person of the same or different sex.

### 2.2. Measures

The Sociodemographic and Sexual History questionnaire includes questions about gender, age, level of education, nationality, relationship status (*not involved in a steady relationship* or *in a steady relationship*), number of sexual partners, and age of first sexual relations.

The Kinsey Scale [29] was used to assess with whom the study participants have sexual relations. This scale uses eight response options for sexual practices, from “exclusively heterosexuals” (option 1) to “exclusively homosexuals” (option 7). An eighth option was included to account for asexuality. This study specifically included participants who chose either option 1 or option 7. Out of those who selected different response options (2 to 6 or 8), the gender distribution revealed 292 men and 342 women.

In terms of the rewards/costs checklist [30], we employed the Spanish adaptation [25,31]. This checklist comprises 58 items to assess different sexual exchange types. According to their current relationship, the participants value each item as a sexual reward, sexual cost, both (a reward and cost), or neither (no reward and no cost). For example, Exchange 12: “Oral sex: extent to which your partner stimulates you” could be considered a sexual reward if the person considers said behavior to be pleasant, or it could be considered a sexual cost if this behavior does not occur, and the person would like to receive oral sex from their partner. Likewise, it could be considered a reward and a cost if, for example, it is positive because the person enjoys it, but it is also negative at the same time because the person considers that it should be a more frequent practice. Lastly, this exchange may not be relevant to the person, so they may not respond. It is important to mention that some of the items were modified so that people who have a relationship with a person of the same sex could answer. For example, “Ability/inability to have a child” was replaced with “Ability/inability to have a child (either adopted or biological)” or “Method of protection (from sexually transmitted infections and/or pregnancy) used by you and your partner” to “Method of protection (from sexually transmitted infections) used by you and your partner”. In the present research, a sexual exchange was considered a reward if the participants indicated it as a reward, a cost if the participants indicated it as a cost, and a reward and a cost if the participants indicated the exchange as a reward and a cost. The rewards/costs checklist is not a measure of a latent construct but rather a list of sexual exchanges. Therefore, it lacks evidence of internal structure validity and internal consistency reliability.

### 2.3. Procedure

Instruments were administered in both the traditional pencil-and-paper format and the online format. The participants who completed the questionnaires in the paper-and-pencil format were contacted at educational and social centers and in public places. In addition, the gay participants were also contacted through lesbian, gay, bisexual, and transgender associations. The evaluator delivered and collected the questionnaires. After answering the scales, the participants returned them to the evaluators in a sealed envelope to guarantee maximum confidentiality. For the online version, the questionnaire battery was created using the Limesurvey™ software (Limesurvey GmbH, Hamburg, Germany) and distributed via virtual platforms (Facebook™, Twitter™, WhatsApp™, and e-mail). To access the questionnaire battery, the participants had to confirm a random sum. For both formats, all the participants voluntarily completed the questionnaires, and their anonymity and the confidentiality of their answers were always guaranteed. All those who decided to contribute to this study had to read an informed consent form and indicate their agreement to participate in the study by selecting the option confirming their understanding and willingness to take part. The estimated time to complete the questionnaires was 10 min.

### 2.4. Data Analyses

A comprehensive data analysis was conducted using IBM SPSS Statistics v. 29, a predictive analytics software, to examine the characteristics of the sample. Descriptive analyses were performed to assess the frequencies of exchanges categorized as rewards, costs, and rewards and costs, with the sample stratified by gender and sexual orientation. These frequencies were then ranked in descending order based on the observed percentage values within each group. Subsequent analyses focused on comparing rewards, costs, and rewards and costs among the four groups, building upon the ranking results. Custom tables were employed to facilitate this comparison, and a chi-square analysis was conducted to assess the differences between the groups for each category of exchange (reward, cost, and reward and cost). To address the potential increase in type I errors due to multiple testing, the Benjamini–Hochberg correction method was applied [32]. This correction method is deemed particularly effective when tests are either independent or only weakly dependent. In the specific case of the analysis of the RCC measure across several subgroups, it should be noted that these subgroups are not interrelated. Consequently, the results of one test are independent of the results of others within this context. Corresponding values and *p*-values for each category were obtained through a chi-square analysis. Subsequently, the obtained *p*-values were subjected to the Benjamini–Hochberg correction to assess specific group differences. For the calculation of the *p*-value threshold in conjunction with the Benjamini–Hochberg correction method, the analysis was conducted using SPSS v. 29 software, with the custom tables option selected and the Benjamini–Hochberg correction method chosen within the test statistics section. This approach for independent multiple comparisons resulted in a significance level set at 0.05. Additionally, the strength of association between variables was quantified using Cramer’s V coefficient [33]. To interpret the magnitude of associations, predefined ranges were considered, as outlined by Rea and Parker [34], providing a framework to evaluate the intensity of connections between variables.

## 3. Results

### 3.1. Sexual Rewards, Costs, and Rewards and Costs

The results indicate notable patterns in reported sexual exchanges as rewards across different subgroups. Among heterosexual men, the top five exchanges with the highest percentages were identified as follows: item 1 (96.6%), item 57 (95.2%), item 41 (95.1%), item 47 (95%), and item 32 (94.6%). Notably, item 1 pertains to the “Level of affection you and your partner express during sexual activities”. Conversely, among heterosexual women, the most prevalent exchanges were item 32 (94.6%), item 35 and item 14 (both at 94.4%), item 42 (93.4%), item 8 (93.2%), and item 45 (92.5%). The leading exchange (item 32) relates to “How comfortable you and your partner feel with each other”. In the case of gay men, the top five exchanges were item 32 (94.2%), item 14 (93.5%), item 1 (91.9%), item 42 (91.7%), and item 41 (91.3%). Remarkably, the item with the highest percentage among gay men aligned with the one reported by heterosexual women. Finally, within the lesbian subsample, the most prevalent exchanges were item 32 (96.7%), item 16 (95.2%), item 23 (95%), item 14 (94.7%), and items 29 and 35 (both at 94.4%). Appendix A includes the percentages of sexual exchanges as a sexual reward reported by the four subsamples.

Appendix B presents a detailed breakdown of the reported percentages of sexual exchanges as sexual costs within each subgroup. Among heterosexual men, the exchanges associated with the highest percentage values across all 58 items were item 27 (62.7%), item 26 (56%), item 5 (50%), item 44 (49.4%), and item 15 (49.2%). Notably, the exchange with the highest percentage (item 27) pertains to “Having sex when your partner is not in the mood”. Similarly, among heterosexual women, the costliest exchanges were item 26 (68.8%), item 27 (62.1%), item 15 (58%), item 34 (55.3%), and item 5 (54.4%). In this case, the primary costly exchange (item 26) refers to “Having sex when you’re not in the mood”. Comparatively, gay men identified the costliest items as item 27 (70.9%), item 26 (61.4%), item 21 (60.6%), item 15 (55.4%), and item 5 (53.4%). The exchange item with the highest percentage among gay men aligns with the one reported by heterosexual men, as mentioned previously. Moreover, in the lesbian subgroup, the most prevalent exchanges were 27 (63.8%), item 26 (61.6%), item 15 (60.9%), item 52 (53.8%), and item 30 (53.3%).

Additionally, considering the exchanges that were reported as a reward and cost (which is referred to as “reward and cost”), among heterosexual men, the highest percentages were associated with item 10 (13.6%), item 4 (12.5%), item 36 (11.7%), item 17 (10.5%), and item 9 (9.2%). Notably, item 10, which refers to “Frequency of sexual activities”, was the primary item selected as reward and cost. For heterosexual women, the highest percentages were reported for item 10 (13.1%), item 4 (12.6%), item 17 (9.9%), items 36 and 48 (9.6%), and item 6 (8.3%). Again, item 10 was the number one exchange selected. Among gay men, the most rewarding and costly items were item 58 (12.5%), item 17 (11.8%), item 51 (11.3%), item 4 (10.8%), and items 10 and 22 (10.5%). Markedly, item 58 was the highest exchange, and it refers to the “Extent to which you and your partner are sexually compatible” (i.e., well matched in terms of your sexual likes/dislikes). Finally, in the lesbian subgroup, the highest percentages were reported for item 10 (14.1%), item 17 (12.8%), item 4 (11.6%), item 6 (9.8%), and item 36 (9.4%). Item 10, once again, was the most selected in this subsample. Appendix C includes the percentages of sexual exchanges as a sexual reward and cost reported by the four subsamples.

### 3.2. Differences between Groups

Table 2 presents a comparative analysis encompassing four distinct groups, namely heterosexual men, heterosexual women, gay men, and lesbians. The examination focuses on the percentages associated with the 58 sexual exchanges in terms of sexual rewards, costs, and rewards and costs. Moreover, in relation to the magnitude of the associations, four exchanges presented moderate magnitudes, as established by Rea and Parker [34]. These were items 21, 30, 31, and 54. This suggests that in these items, there is some relationship between the variables, but it is not extremely strong. The rest of the exchanges showed lower magnitudes. Among these 58 sexual exchange items, statistical analysis reveals significant differences in percentage values across the four groups for 38 of them. Out of these 38 items, heterosexual men exhibit the highest percentage values in 15 instances related to sexual rewards, followed by lesbians with 12, heterosexual women with 8, and gay men with 2.

In terms of costs, among the 38 sexual exchanges exhibiting significant differences, heterosexual women recorded the highest percentage value in 10 items, followed by gay men with 8, and finally, both heterosexual men and lesbians with 6 each.

Finally, in the sexual exchanges categorized as rewards and costs, the group of gay men displays the highest percentage value in 10 items, while heterosexual women rank second with 3 items. Heterosexual men and lesbians each exhibit one item.

## 4. Discussion

According to the IEMSS, sexual satisfaction is an affective response resulting from the subjective evaluation of positive and negative aspects experienced during sexual intercourse [16]. Sexual relations in the couple context can present advantages and disadvantages for the people who participate in them. Therefore, this research aims to contribute to a better understanding of sexual satisfaction by assessing sexual exchanges as rewards (i.e., pleasant or positive), costs (i.e., negative or unpleasant), or rewards and costs in a Spanish-speaking adult sample. For this, a ranking was conducted of the positive (rewards), negative (costs), and positive and negative (rewards and costs) exchanges reported by men and women in a sexual relationship with people of different or the same sex. Likewise, possible differences in the 58 sexual exchanges, such as rewards and costs, depending on gender and sexual orientation, were examined.

In general, most reward exchanges in the four groups were generally related to emotional and physical aspects within the context of sexual and partner relationships. This could be the result of a dynamic interpersonal process based on intimacy. This intimacy within couple relationships can be characterized by openness (e.g., items 41 and 42), affection (e.g., item 1 and item 14), support and union and silent company (e.g., item 32), and sexual intimacy (e.g., item 29). Previous research has shown that intimacy is directly associated with sexual satisfaction [35,36,37] and with relationship satisfaction [38,39] in both heterosexual and gay populations. According to all this, intimacy seems to play an important role in sexual relationships and relationships regardless of people’s gender and/or sexual orientation.

More specifically, for rewards, it was found that the most beneficial exchange for gay men, heterosexual women, and lesbians was item 32 (“How comfortable you and your partner feel with each other”); this item also appears among the highest-scored in heterosexual men. This exchange refers to the level of comfortability that the person and his/her partner feel with each other. This reward does not seem to depend on the gender or sexual orientation of the respondents. In heterosexual men, item 1, referring to the level of affection that the person and their partner express during sexual activity, was considered the most rewarding exchange. This item was also one of the most valued by gay men (third place). However, although heterosexual and lesbian women rate this exchange among the ten most valued as rewards, it seems not to be as relevant for them. This may be due to socialization based on stereotyped gender roles that, in the case of women, unlike men, are more oriented towards the well-being of others [20,40,41]. In this sense, the level of affection achieved in a sexual relationship can be considered by the group of men as an achievement, while, in the case of women, there are rewards more related to the well-being of the couple, for example, their frequency of orgasms (item 8), being able to please her (item 45), and having fun with it (item 16).

In terms of costs, four groups agree on three exchanges of the costliest exchanges. These refer to having sex when you are not in the mood, having sex when your partner is not in the mood, and feeling physical discomfort/pain during/after sex (items 26, 27, and 15, respectively). The consideration of these exchanges as costs reveals social progress in the sexual field, in terms of sexual and reproductive rights, related to the freedom to decide on sexuality and the free exercise of it in a pleasant and safe way [42,43,44,45]. On the other hand, item 5, referring to the use of sex toys, appears as a cost in heterosexual women and in heterosexual and gay men, but to a lesser extent in lesbians. Considering the use of sex toys by lesbians as less expensive may be related to item 16 of fun with the partner chosen as the second most beneficial exchange by them. This result could also be because lesbians, compared to heterosexual women, use sex toys more frequently [46].

The four groups place four exchanges in the top five of the rewards and costs ranking (except for item 36, which ranks 7th in gay men). The content of these exchanges refers to the variety (item 4) and frequency (item 10) of sexual activities, which initiates the sexual activity (item 17), and the time dedicated to it (item 36). These aspects of sexual activity in the context of a couple would be valued positively, but in turn, the respondents would like them to be different (e.g., more/less diversity, more/less frequency, or more/less time spent). It should be noted that the most valued exchange as rewards and cost by gay men is item 58, related to the degree of sexual compatibility between the couple; that is if they coincide in terms of sexual likes/dislikes. In this line, this group also reports item 51, related to participation in sexual activities that are pleasurable but displeasing to the partner among the exchanges, most scored as rewards and cost. It seems that, in these couples, the compatibility between both members, despite being valued as a reward, sometimes does not meet the expectations created, so it would also be valued as a cost.

When comparing rewards and costs, the heterosexual men group reported the most rewards compared to the other groups, while heterosexual women reported more costs. As previously mentioned, gender socialization may be influencing, such that heterosexual women in this study might be putting their partner’s sexual needs before their own [47]. This result may be related to sexual indulgence. This term has been defined as consensual but unwanted sexual activity [48], which is carried out to fulfill the sexual desire of a sexually interested partner. According to Impett and Peplau [49], this complacency can be explained from three perspectives: gender, motivation, and relationship maintenance. From a gender perspective, sexual complacency may be due to an existing asymmetry of power in couples and is more frequent between men and women. Likewise, it should be noted that the women who support a double sexual standard present less autonomy, sexual assertiveness, and sexual satisfaction [50,51,52,53]. On the other hand, it is observed that gay men are the ones who report more exchanges as reward and cost compared to the rest of the groups. The content of these most scored exchanges refers to affection (item 1) and intimacy with the couple (item 9), fun in the sexual relationship (item 16), exclusivity (item 35), if they have broken stereotypes of gender (item 39) and couple compatibility, concern about sexually transmitted infections and partner attraction (items 58, 52, 57, respectively). These exchanges are considered by gay men as rewards, but at the same time, they are being evaluated as sexual costs. This may be due to a “dissatisfaction” based on the desire to be able to achieve greater satisfaction in these exchanges; that is, they are already rewards for them, but they would like to experience them in another way (e.g., more frequency, greater intensity, etc.). In this sense, according to a previous study [26], in gay men, the positive perception of sexual exchanges must be high enough for them to feel satisfied with their relationship. In the same way, it can happen with the valuation of sexual exchanges as sexual costs; perhaps this negative perception of sexual exchanges should be high enough to be considered a cost and that it affects sexual satisfaction. All this can also explain why these exchanges are valued as rewards and costs by these people at the same time.

These findings carry significant implications as they reveal patterns and differences in the occurrence of exchanges perceived as rewards, costs, or a combination of both among the analyzed groups. Given that this checklist is grounded in a well-established theoretical model of sexual satisfaction, we can identify and gain a deeper understanding of the aspects (represented by the list of exchanges) that positively or negatively influence sexual satisfaction according to the theoretical model. Furthermore, these findings can inform programs and approaches aimed at enhancing the quality of sexual experiences.

Finally, it is worth mentioning that, in terms of sociodemographic information, men across different age ranges and sexual orientations had a significantly higher mean age than women. This aligns with findings from a recent study by Muñoz-García et al. [54] involving a Hispanic population with varied sexual orientations. Concerning the number of sexual partners, men consistently reported a significantly higher count than women, irrespective of sexual orientation. This pattern could be linked with factors linked to sexual sensation seeking, as supported by prior research highlighting the relation between sexual sensation seeking and an increased number of sexual partners [55,56]. Moreover, existing studies suggest that heterosexual men are more likely to exhibit sexual sensation-seeking tendencies compared to their female counterparts [57,58,59], and individuals belonging to sexual minority groups demonstrate a heightened inclination toward sexual sensation-seeking in comparison to heterosexual individuals [58,59,60]. In light of those findings, it could be contended that the observed higher number of sexual partners among male participants in this study is intricately tied to their proclivity for seeking sensory sexual experiences. Lastly, concerning the level of education, significant gender disparities were identified solely within the heterosexual group. Despite a lower representation of women in the first two education levels, a higher proportion of women demonstrated higher education compared to men. That reflects evolving social norms, indicating a perceptible change in women’s attitudes towards higher education. Women now enjoy increased support and accessibility to educational pathways, overcoming previous limitations imposed by cultural or social norms in Hispanic countries.

The interplay of these characteristics could potentially impact the assessment of specific sexual exchanges, such as engaging in sexual activity when one’s partner is not in the mood or evaluating the level of affection during intimate moments. Given that engaging in sexual sensation seeking reflects a readiness to pursue novel and exhilarating experiences, potentially contributing to an augmented number of sexual partners, men (compared to women), particularly those of a certain age bracket reporting a higher number of sexual partners, might be inclined towards actively seeking diversity and freshness in their sexual encounters with their romantic partner to attain a specific level of affection, regardless of their sexual orientation. Conversely, in terms of perceived costs, exploring new experiences might be a drawback for men if they sense their partner is not in the mood. Nevertheless, a more in-depth investigation is warranted to comprehensively explore these constructs.

The present study is not without its limitations. First, the results cannot be generalized because intentional non-probabilistic sampling was used. In addition, as most of the participants had university studies, it would be interesting for future research to find out if the level of education is related in any way to the types of rewards and costs reported by both men and women. Another limitation is that there is no evidence available regarding the invariance of the scale used by gender and sexual orientation, so it is possible that it is not neutral. In addition, we did not compare whether there were differences between participants who completed the paper-and-pencil versus online questionnaires. Finally, future research with clinical samples is recommended (e.g., sexual dysfunction, general sexual dissatisfaction, and sexual dissatisfaction with a partner), in which the relationship of sexual exchanges identified as rewards and/or costs with other variables such as sexual attitudes attachment, or relationship dynamics are analyzed. We also propose that future research should consider the comparison level against previous sexual activity and would feature questions like “Compared to your past experiences with oral sex, how rewarding is oral sex now with your partner?”, and the comparison level for alternatives and might feature questions like “How likely would oral sex with other possible partners be better than it is with your current partner?”. These questions could also be classified by gender and sexual orientation. For example, “How likely is it that you would have more frequent sexual activity if your partner was of a different sexual orientation than your current partner?” or “How likely is it that you would have less frequent sexual activity if your partner was of a different sexual orientation?”.

## 5. Conclusions

This research extends knowledge on (1) one of the few theoretical models on sexual satisfaction that exists, (2) aspects related to sexual relations in the Spanish-speaking population, and (3) dimensions that contribute to sexual satisfaction in people with different sexual orientations. Although there are differences in the number of rewards and costs between the examined groups, all the participants generally reported similarities in the type of sexual rewards and costs. It should be noted that the emotional/interpersonal and physical aspects of sexual relations and the couple’s relationship appeared in the two most reported sexual exchanges by the four groups in both the form of rewards and costs. This reiterates that these aspects play a crucial role in sexual satisfaction. Finally, it was evident that the groups made up of heterosexual men and heterosexual women were those with the most rewards and costs, respectively.

## Figures and Tables

**Table 1 healthcare-12-00170-t001:** The participants’ sociodemographic characteristics (*N* = 1996).

Variables	Heterosexual			Homosexual		
	Men *n* = 538	Women *n* = 550		Men *n* = 504	Women *n* = 404	
Age rank (years)	18–74	18–74		18–58	18–58	
	M (SD)	M (SD)	t	M (SD)	M (SD)	t
Age (years)	34.52 (12.78)	31.10 (11.85)	4.57 ***	31.05 (9.35)	28.67 (8.40)	3.98 ***
Number of sexual partners	6.04 (9.56)	3.80 (4.90)	4.87 ***	39.18 (82.97)	7.45 (9.44)	7.64 ***
First sexual relation (years)	17.69 (2.86)	17.92 (2.94)	−1.30	17.14 (3.99)	17.22 (2.99)	−0.33
Level of education No studies	*n* (%) 8 (1.5)	*n* (%) 4 (0.7)	*χ* ^2^	*n* (%) 0	*n* (%) 0	*χ* ^2^
Primary education	63 (11.8)	48 (8.7)	12.36 ***	7 (1.4)	8 (2)	0.71
Secondary education	181 (34)	152 (27.6)		107 (21.2)	90 (22.2)	
Tertiary education	281 (52.7)	346 (62.9)		390 (77.4)	306 (75.7)	

*Note.* M: mean; SD: standard deviation. *** *p* < 0.001.

**Table 2 healthcare-12-00170-t002:** Comparison of percentages between heterosexual men, women, gay men, and lesbians on sexual rewards, costs, and rewards and costs.

Sexual Exchanges		Heterosexual Men (A)	Heterosexual Women (B)	Gay Men (C)	Lesbians (D)	*X* ^2^	*p*	*V*
Level of affection you and your partner express during sexual activities	Reward	96.6% _B_ ^1^_, C_	92.2%	91.9%	93.5%	15.43	0.017	0.063
	Cost	1.9%	4.4%	4%	2.2%			
	Rewards and Costs	1.5%	3.3%	4.2% _A_	4.2% _A_			
2. Degree of emotional intimacy (feeling close, sharing feelings)	Reward	92.5%	90%	90.2%	94.3%	14.63	0.023	0.061
	Cost	4.4%	6.6% _D_	4.2%	2.8%			
	Rewards and Costs	3.1%	3.3%	5.6%	3%			
3. Extent to which you and your partner communicate about sex	Reward	75.9%	82.1% _A, C_	75.8%	79.3%	15.31	0.018	0.063
	Cost	17.5%	15.1%	16.3%	15.5%			
	Rewards and Costs	6.6% _B_	2.9%	7.9% _B_	5.2%			
4. Variety in sexual activities, locations, times	Reward	57%	62.4%	59.8%	59.5%	5.13	0.526	0.037
	Cost	30.5%	25%	29.3%	28.9%			
	Rewards and Costs	12.5%	12.6%	10.8%	11.6%			
5. Extent to which you and your partner use sex toys	Reward	43.8%	39.8%	41.6%	54.5% _A, B, C_	27.91	<0.001	0.092
	Cost	50% _D_	54.4% _D_	53.4% _D_	37.7%			
	Rewards and Costs	6.3%	5.8%	5%	7.9%			
6. Sexual activities you and your partner engage in to arouse each other	Reward	76.4%	77.6%	74.9%	79.3%	5.38	0.496	0.037
	Cost	15%	14.1%	15.9%	11%			
	Rewards and Costs	8.6%	8.3%	9.2%	9.8%			
7. How often you experience orgasm (climax)	Reward	90.3% _B, C, D_	78.7%	85.2% _C_	82.5%	28.62	<0.001	0.086
	Cost	5.7%	13.2% _A, C_	8.5%	10% _A_			
	Rewards and Costs	4%	8.1% _A_	6.3%	7.5%			
8. How often your partner experiences orgasm (climax)	Reward	80.7%	93.2% _A, C, D_	83.6%	87.3% _A_	38.66	<0.001	0.100
	Cost	12.4% _B, D_	4%	9.9% _B_	7.5% _B_			
	Rewards and Costs	6.9% _B_	2.8%	6.5% _B_	5.3%			
9. Extent to which you and your partner engage in intimate activities (e.g., talking, cuddling) after sex	Reward	83.9%	81.3%	82.7%	92.8% _A, B, C_	27.25	<0.001	0.084
	Cost	11.1% _D_	12.6% _D_	12% _D_	4.5%			
	Rewards and Costs	5%	6.1%	5.2%	2.7%			
10. Frequency of sexual activities	Reward	57.8%	63.9%	58.3%	57.3%	11.97	0.062	0.056
	Cost	28.7%	23%	31.2%	28.5%			
	Rewards and Costs	13.6%	13.1%	10.5%	14.1%			
11. How much privacy you and your partner have for sex	Reward	80.1%	79%	85.3%	84.4%	11.58	0.072	0.055
	Cost	14.9%	15.8%	9.9%	10.8%			
	Rewards and Costs	5%	5.3%	4.8%	4.8%			
12. Oral sex: extent to which your partner stimulates you	Reward	70.4%	72%	76%	76.3%	9.43	0.151	0.050
	Cost	23.7%	21.2%	17.1%	17.2%			
	Rewards and Costs	6%	6.8%	6.9%	6.6%			
13. Oral sex: extent to which you stimulate your partner	Reward	73.1%	72.9%	80.9% _A, B_	77.2%	13.89	0.031	0.061
	Cost	19.3% _C_	20.8% _C_	13.1%	16.3%			
	Rewards and Costs	7.6%	6.4%	6%	6.5%			
14. Physical sensations from touching, caressing, hugging	Reward	90.5%	94.4% _A_	93.5%	94.7% _A_	13.34	0.038	0.059
	Cost	6% _D_	3.9%	3.9%	25%			
	Rewards and Costs	3.5%	1.7%	2.7%	3.3%			
15. Feelings of physical discomfort or pain during/after sex	Reward	47.6% _B, D_	36.6%	40.1%	33.4%	17.51	0.008	0.076
	Cost	49.2%	58% _A_	55.4%	60.9% _A_			
	Rewards and Costs	3.2%	5.4%	4.5%	5.6%			
16. How much fun you and your partner experience during sexual interactions	Reward	92.9% _C_	91.1%	87.5%	95.2% _B, C_	27.62	<0.001	0.085
	Cost	5.1% _D_	5.6% _D_	5.8% _D_	1.8%			
	Rewards and Costs	2%	3.3%	6.7% _A, B, D_	3%			
17. Who initiates sexual activities	Reward	67%	75.2% _A, C_	67.1%	72.2%	16.03	0.014	0.068
	Cost	22.5% _B, D_	14.9%	21.1% _B, D_	15%			
	Rewards and Costs	10.5%	9.9%	11.8%	12.8%			
18. Extent to which you feel stressed/relaxed during sexual activities	Reward	87.3% _C_	83.4%	78%	82%	18.27	0.006	0.071
	Cost	8.6%	12.1%	15.4% _A_	10.7%			
	Rewards and Costs	4.1%	4.5%	6.6%	7.3%			
19. Extent to which you and your partner express enjoyment about your sexual interactions	Reward	91.2%	92.1%	89%	93.5%	6.89	0.330	0.042
	Cost	4.7%	4.9%	5.9%	3.5%			
	Rewards and Costs	4.1%	3%	5.1%	3%			
20. Extent to which you and your partner communicate your sexual likes and dislikes to each other	Reward	78.2%	80.4% _C_	74.3%	85.1% _A, C_	16.83	0.010	0.066
	Cost	15.4% _D_	13.7%	17.8% _D_	9.6%			
	Rewards and Costs	6.4%	5.9%	7.9%	5.3%			
21. Ability/inability to conceive a child	Reward	74.4% _C, D_	68% _C, D_	31.3%	43.9% _C_	199.98	<0.001	0.259
	Cost	19.6%	28.1% _A_	60.6% _A, B, D_	49.4% _A, B_			
	Rewards and Costs	5.9%	3.9%	8.1%	6.7%			
22. Extent to which you and your partner engage in role-playing or act out fantasies	Reward	55.9%	56%	52.6%	55.7%	9.58	0.143	0.052
	Cost	38.1%	37.7%	37%	38%			
	Rewards and Costs	6%	6.3%	10.5%	6.3%			
23. How you feel about yourself during/after engaging in sexual activities with your partner	Reward	93.6% _C_	92.4%	89%	95% _C_	19.83	0.003	0.072
	Cost	4.1%	4.5%	4.5%	2.8%			
	Rewards and Costs	2.3%	3%	6.5% _A, B, D_	2.3%			
24. Extent to which your partner shows consideration for your wants/needs/feelings	Reward	82% _C_	84.6% _C_	75.8%	89.4% _A, B, C_	33.68	<0.001	0.093
	Cost	12.5% _D_	9.7% _D_	16.1% _B, D_	5.5%			
	Rewards and Costs	5.5%	5.7%	8.1%	5%			
25. How your partner treats you (verbally and physically) when you have sex	Reward	91.5%	90.1%	88.4%	93%	9.28	0.158	0.049
	Cost	5.5%	7.3%	7.1%	3.8%			
	Rewards and Costs	3%	2.7%	4.5%	3.3%			
26. Having sex when you’re not in the mood	Reward	38.5% _B_	24.9%	33.3% _B_	34.7% _B_	21.41	0.002	0.080
	Cost	56%	68.8% _A_	61.4%	61.6%			
	Rewards and Costs	5.5%	6.3%	5.3%	3.7%			
27. Having sex when your partner is not in the mood	Reward	30.8%	32.1%	24.4%	31.4%	10.50	0.105	0.057
	Cost	62.7%	62.1%	70.9%	63.8%			
	Rewards and Costs	6.5%	5.8%	4.7%	4.8%			
28. Extent to which you let your guard down with your partner	Reward	93.1%	90.4%	89.6%	93.7%	14.06	0.029	0.061
	Cost	5.3%	7.6%	6.3%	3.8%			
	Rewards and Costs	1.6%	1.9%	4.1%	2.5%			
29. Extent to which your partner lets their guard down with you	Reward	87.7%	90.5%	87.9%	94.4% _A, C_	16.28	0.012	0.066
	Cost	9.3% _D_	7.3% _D_	8% _D_	3.8%			
	Rewards and Costs	3%	2.2%	4.1%	1.8%			
30. Method of protection (from sexually transmitted infections and/or pregnancy) used by you and your partner	Reward	77.4% _C, D_	80.8% _C, D_	60.9% _D_	43.5%	178.43	<0.001	0.233
	Cost	17.3%	13.8%	32.7% _A, B_	53.3% _A, B, C_			
	Rewards and Costs	5.3%	5.3%	6.4%	3.3%			
31. Extent to which you and your partner discuss and use protection (from sexually transmitted diseases and/or pregnancy)	Reward	79.3% _C, D_	81.4% _C, D_	68.3% _D_	45.9%	144.41	<0.001	0.210
	Cost	16.6%	16.3%	27.1% _A, B_	50.5% _A, B, C_			
	Rewards and Costs	4.1%	2.3%	4.7%	3.6%			
32. How comfortable you and your partner are with each other	Reward	94.6%	94.6%	94.2%	96.7%	8.18	0.225	0.046
	Cost	3.9%	3.1%	3%	2.8%			
	Rewards and Costs	1.5%	2.2%	2.8%	0.5%			
33. Extent to which/way in which your partner influences you to engage in sexual activity	Reward	76.7%	76.4%	75.8%	83.1%	12.17	0.058	0.058
	Cost	17.9%	18%	16.4%	13.5%			
	Rewards and Costs	5.4%	5.6%	7.8% _D_	3.3%			
34. Extent to which you and your partner argue after engaging in sexual activity	Reward	51.5%	42.5%	48.5%	45.5%	11.19	0.083	0.062
	Cost	45.5%	55.3%	47%	51%			
	Rewards and Costs	3%	2.2%	4.4%	3.6%			
35. Extent to which you and your partner are/are not sexually exclusive (i.e., have sex only with each other)	Reward	87.4% _C_	94.4% _A, C_	71.6%	94.4% _A, C_	143.25	<0.001	0.196
	Cost	7.7% _B, D_	4.5%	19.8% _A, B, D_	3.1%			
	Rewards and Costs	4.9% _B_	1.2%	8.5% _A, B, D_	2.5%			
36. How much time you and your partner spend engaging in sexual activities	Reward	60.8%	69.6% _A_	62.6%	67.5%	12.32	0.055	0.057
	Cost	27.4%	20.9%	27.7%	23.1%			
	Rewards and Costs	11.7%	9.6%	9.7%	9.4%			
37. How easy it is for you to have an orgasm (climax)	Reward	89.7% _B, C, D_	75.5%	81.9% _B, D_	73.7%	50.11	<0.001	0.114
	Cost	7.6%	19.2% _A, C_	12.8% _A_	20.3% _A, C_			
	Rewards and Costs	2.7%	5.4%	5.3%	6.1%			
38. How easy it is for your partner to have an orgasm (climax)	Reward	76%	90.8% _A, C, D_	79.4%	83% _A_	44.73	<0.001	0.109
	Cost	17.1% _B_	5.2%	14.1% _B_	12.7% _B_			
	Rewards and Costs	6.9%	4%	6.5%	4.3%			
39. Extent to which your sexual relationship with your partner reflects or breaks down stereotypical gender roles (the way women and men are expected to behave sexually)	Reward	76.4% _C_	71.8%	66.3%	78.8% _C_	36.24	<0.001	0.105
	Cost	20.2% _D_	23.3% _D_	23.4% _D_	13.5%			
	Rewards and Costs	3.3%	4.9%	10.3% _A, B_	7.7% _A_			
40. How your partner responds to your initiation of sexual activity	Reward	73.4%	90.3% _A, C, D_	77.5%	83.6% _A, C_	58.07	<0.001	0.123
	Cost	20.3% _B, D_	6%	16.4% _B, D_	10.8% _B_			
	Rewards and Costs	6.3%	3.7%	6.1%	5.5%			
41. Being naked in front of your partner	Reward	95.1% _B, C, D_	85.6%	91.3% _C_	89.6%	29.323	<0.001	0.087
	Cost	3.8%	10.9% _A, C_	5.7%	7.1%			
	Rewards and Costs	1.2%	3.4%	3%	3.3%			
42. Your partner being naked in front of you	Reward	89.2%	93.4%	91.7%	92.4%	6.99	0.321	0.043
	Cost	8.8%	5%	6.5%	5.8%			
	Rewards and Costs	2%	1.5%	1.8%	1.8%			
43. Extent to which your partner talks to other people about your sex life	Reward	56.3% _B_	44.1%	49.9%	48.4%	18.51	0.005	0.075
	Cost	39.9%	50.5% _A_	42.5%	45.2%			
	Rewards and Costs	3.8%	5.5%	7.7%	6.4%			
44. Extent to which you and your partner read/watch sexually explicit material (e.g., erotic stories, pornographic videos)	Reward	43.2%	41.1%	48.8%	44.9%	10.19	0.117	0.056
	Cost	49.4%	53.9%	44.1%	47.2%			
	Rewards and Costs	7.4%	5%	7.2%	7.9%			
45. Pleasing/trying to please your partner sexually	Reward	91.8% _C_	92.5% _C_	87.1%	92.7% _C_	14.96	0.021	0.062
	Cost	4.9%	5%	6.7%	4.1%			
	Rewards and Costs	3.3%	2.5%	6.3% _B_	3.3%			
46. Extent to which sexual interactions with your partner make you feel secure in the relationship	Reward	90.5%	89.5%	88.1%	89.7%	2.51	0.867	0.026
	Cost	5.9%	6.6%	7.2%	5.5%			
	Rewards and Costs	3.6%	3.9%	4.7%	4.8%			
47. Extent to which you get sexually aroused	Reward	95% _B, C_	90.3%	87.1%	92.2% _C_	21.71	0.001	0.075
	Cost	3.1%	6.5% _A_	7.4% _A_	5%			
	Rewards and Costs	1.9%	3.2%	5.4% _A_	2.8%			
48. Amount of spontaneity in your sex life	Reward	68.7%	69.9%	69%	74.8%	7.64	0.266	0.045
	Cost	22.1%	20.5%	22.8%	16%			
	Rewards and Costs	9.2%	9.6%	8.3%	9.2%			
49. Extent of control you feel during/after sexual activity	Reward	89.3%	87%	88%	89.4%	4.04	0.671	0.034
	Cost	8.5%	9.8%	8.8%	6.8%			
	Rewards and Costs	2.3%	3.2%	3.2%	3.8%			
50. Extent to which you engage in sexual activities that you dislike but your partner enjoys	Reward	57.4% _B_	43.5%	50.7%	55.7% _B_	26.69	<0.001	0.094
	Cost	36.7%	48.9% _A_	43.8%	42.5%			
	Rewards and Costs	5.8% _D_	7.6% _D_	5.5% _D_	1.8%			
51. Extent to which you engage in sexual activities that you enjoy but your partner dislikes	Reward	50.6% _C_	44.2%	40.8%	47.9%	16.91	0.010	0.074
	Cost	42.4%	49.7%	47.9%	45.9%			
	Rewards and Costs	7%	6.1%	11.3%	6.2%			
52. Worry that you or your partner will obtain a sexually transmitted infection from each other	Reward	57.9% _B, C, D_	45.7%	48.5%	42%	31.09	<0.001	0.102
	Cost	38.7%	51.9% _A_	44.7%	53.8% _A, C_			
	Rewards and Costs	3.4%	2.4%	6.8% _B_	4.3%			
53. How confident you feel in terms of your ability to please your partner sexually	Reward	88.3%	88.8%	83.9%	88.1%	10.33	0.111	0.052
	Cost	5.7%	7.1%	9.6%	7.6%			
	Rewards and Costs	6.1%	4%	6.5%	4.3%			
54. Extent to which you and your partner engage in anal sex/anal play	Reward	47.4%	51.5%	81.6% _A, B, D_	52%	157.24	<0.001	0.214
	Cost	45.7% _C_	43.4% _C_	13.2%	40.8% _C_			
	Rewards and Costs	7%	5.1%	5.3%	7.2%			
55. Your partner’s ability to please you sexually	Reward	88.7% _C_	88.5% _C_	81.4%	87.6% _C_	19.21	0.004	0.071
	Cost	7%	5.6%	11.9% _A, B, D_	7.4%			
	Rewards and Costs	4.3%	6%	6.7%	5.1%			
56. Extent to which you think your partner is physically attracted to/sexually desires you	Reward	87.1%	89.3%	84.9%	88.3%	9.90	0.129	0.051
	Cost	8.8%	6.9%	8.4%	5.7%			
	Rewards and Costs	4.1%	3.8%	6.6%	6%			
57. Extent to which you are physically attracted to/sexually desire your partner	Reward	95.2% _B, C_	90.9% _C_	84.9%	92.5% _C_	36.27	<0.001	0.097
	Cost	2.7%	5.5%	7.3% _A_	4.3%			
	Rewards and Costs	2.1%	3.6%	7.7% _A, B, D_	3.3%			
58. Extent to which you and your partner are sexually compatible (i.e., well-matched in terms of your sexual likes/dislikes)	Reward	85.9% _C_	86.9% _C_	79.4%	89.6% _C_	24.24	<0.001	0.079
	Cost	7.1%	6.6%	8.1%	5%			
	Rewards and Costs	7.1%	6.5%	12.5% _A, B, D_	5.5%			

^1^ The subscript letter denotes significant differences between groups, with higher scores for the group, which is represented by the subscript letter.

## Data Availability

The data presented in this study are available on request from the corresponding author.

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
