# Peer review of "Sexual Rewards and Costs in Heterosexual and Gay Hispanic Adults"

_healthcare, 2024, doi:10.3390/healthcare12020170_

Round 1
Reviewer 1 Report
Comments and Suggestions for Authors
Calvillo et al., present evidence as to what people may interpret as sexual reward, cost, or both, by gender and sexual orientation in people currently in a relationship. Their findings provide descriptive data as to which group interpret as sexual reward, cost, or both, as well as significant comparisons showing that there is a gender and sexual orientation effect, yet not an interaction of both factors.
Their work presents several minor mistakes and areas that need revision. The method section lacks some crucial details. However, the most concerning reviews have to do with their results and how they are being communicated. On the descriptive part, the result section is indigestible, meaning there are far too many data points being presented that obnubilate the overall understanding of their findings. On the inference tests performed on those descriptive statistics, the tests performed provide a limited interpretation of a hunch as to what drives the significant differences of a 3x4 chi square. More concerning than that, the MANOVA analyses lack of addressing its assumptions, and the real analysis that needs to be performed, a discriminant descriptive analysis. Much like with ANOVA, the pivotal analysis lies on the interaction of factors (when there is more than one), and main effects are nothings without post-hoc tests or planned comparisons, for the crux of the analysis lies on determining the group differences.
There are a series of matters that need to be addressed:
Minor:
1. The abstract confuses and provides very little understanding of the study rationale, its design, and what the results mean;
2. There is work to be done crafting a better flow between paragraph, as well as how they begin and end;
3. There are far too many mistakes, corrections, and awkward statements throughout the whole document that need revision.
Major:
1. The result section is crowded, lacks of direction given of the so many comparisons, and what little the chi square comparisons give. Whereas I cannot give the authors a suggestion as to how to improve it, I bring this to their attention for these many pages give indeed very little. Perhaps, authors could think of a better way, more condensed (keeping tables 2, 3, 4 as supplementary), that would improve the understanding of their findings, ad what is important beyond ranks. Perhaps a weighted principal component analysis of the categories using their ranks using sample weighting.
https://doi.org/10.1093/mnras/stu2219
2. Authors are required to describe how they calculated the p value threshold given the use of the Benjamini-Hochberg correction method;
3. The MANOVA analysis lacks of properly addressing all its assumptions, plus a proper discriminant descriptive analysis for the significant differences found;
4. Results do not include a measurement invariance by gender nor sexual orientation for the scores of the scale. Whereas the authors recognize this as a limitation—suspiciously in a convenient manner just for the “paper-and-pencil versus online questionnaires” variable—it is but a requirement for the interpretability of their data. What is good to acknowledge a limitation when it is possible to address it? Limitation are those that researchers are either not in control or cannot do anything about it. I do not think performing a measurement invariance analysis is one of them;
Specific feedback for each section can be found below. Finally, given the changes suggested and limitation pointed out in this review, I acknowledge that the discussion will be further reviewed when changes are made, given the impact these will likely have on the study results.
Therefore, I recommend reconsider after major revision (control missing in some experiments).
Title
I wonder why they choose not to use hetero and homosexual in their title.
Abstract
Line 12 = seems repetitive to use “sexual satisfaction” twice.
Line 13 = do the authors mean “balance of sexual reward or cost exchange”? or something like that? Additionally, the jump from the first to the main goal of the study is completely abrupt. Authors need to do something for the problematization of their study to include in between. The way it is, there is no connection between the model, sexual satisfaction, and raking sexual rewards. Finally, what is the rationale of sexual orientation in the study? If none, then why is it part of the study objective?
It is unclear what do authors mean with “interpersonal aspects”, makes it impossible to understand how these, whatever they are, factor in the design. Similarly, “two most reported sexual exchanges”, no idea what it means.
It seems that authors avoid using sexual orientation labels in their title and the middle part of their abstract. Yet, they do use hetero and homosexuals.
It may be my ignorance, but partner-relationship may not be a good keyword. Also, sexual health is a bit too general. Nothing wrong with these, they would just not be my choices.
Overall, I do not understand what the authors did. The abstract needs a thorough review.
Introduction
Line 24 = social interaction is not much an example of social exchange, but perhaps the closest synonym.
Line 27 = “by contemplating those individuals prefer rewarding exchanges over costly ones”. There is something odd about this sentence.
Line 31-32 = that is a wild statement. Do the authors say in the field of sex research, researchers put more attention to individual aspects more than interpersonal? Not only in this reviewer’s limited and biased opinion it would be the opposite, but the reference provided not only does not suggest that, it puts a complete emphasis on the relational aspects of marriage, dating, and other relationships. How is then the “interpersonal context” treated as secondarily compared to studying “individual aspects”?
Line 35-37 = what not make the sentence gender neutral, so as to accommodate the theory to everyone, and not making it about heterosexual people?
Lines 35-40 = I cannot begin to describe how biased and limited is to portrait that about men, that women play no part in the exchange, or that any other genders would not do similar things. I mean, if it is about reward and costs, then we all do it. Isn’t? This comment is not about looks, it is about precision. Telling one side of the story is never accurate. Thus, why not making it gender neutral?
Line 52 = It is unjustified to cut the paragraph there into a new one highly related one. It is basically the continuation of the previous one.
Line 56 = saying something is “very interesting” does not make it interesting.
Line 58 = why do authors do not say anything about this study’ sample, and yet in Line 62 the do say it was Chinese people with no clear reason as to why mention that detail? Same goes for line 66.
The listing of findings in lines 58 to 75 is preceded by the authors claiming that “evaluation of specific sexual exchanges can be very interesting both in clinical practice and in research”. However, neither of the mentioned studies said anything about the clinical practice, leaving reader to guess what the authors mean with “interesting both in clinical practice”. Furthermore, the listing did not follow a rationale, more than just listing findings. This is also true when the researchers single out Spain as something to worth mentioning. Is there something the readers need to know about Spain? Same goes to “Hispanic” people in line 74.
Line 76 = A paragraph begins introducing an idea. Therefore, beginning with “therefore” is awkward. Moreover, I do not think the first line is convincing to introduce “sexual orientation” as the topic of the paragraph.
Line 85 = There is a need to increase perhaps the majority of research topics, would not the authors agree? Therefore, if everything is important, is really anything important? I think the authors were doing a good job so far justifying sexual diverse populations in the study of this theoretical model. Yet, the last line does not bring the paragraph idea home.
The following paragraph is again a continuation of the previous one.
Line 90 = do please remove “pioneering”, or justify such adjective if it is necessary to keep it.
Line 90-92 = Both sentences begin with “therefore”. There is something wrong there.
Line 94 = previously, authors say they will analyse by “sex”, to then say “by men and women”.
Line 97 = in this line’s paragraph, there are three sentences already saying what the objectives/aims will be. I suggest to boil these ideas down, and come up with a much simplify version of this paragraph.
Methods
Line 105-112 = Authors repeat of the information contained in Table 1. I do not think readers need to read it twice to understand it.
Line 118 = I thought those were inclusion criteria. Thus, the ones who met them where accepted. So, if some did not meet them, then why were “inclusion criteria” in the first place?
Congrats in such balanced sample!
Table 1 = if want to denote ranks use en dash (–), and not a minus sign. Also, there is unjustified space in the row “level of education”, at least in comparison to other categories in the “variable” column. Moreover, I suggest to put the category “level of education” and “nationality” with italics. Finally, following today standards, “other” categories should be justified, more so avoided in order to make visible those who are not a majority.
Line 141 = Regarding the Rewards/Costs Checklist, it does not seem intuitive to understand what can be construed as “reward + cost”. Could the authors help readers to understand this better?
Authors are required to describe how they calculated the p value threshold if they used the Benjamini-Hochberg correction method. Furthermore, the BH procedure is valid only when the tests are independent. Therefore, authors need to explain the reader why their tests comply with this assumption, and whenever a test is dependent on another, they need to clarify which correction they used.
Authors do not provide a reference for a standard to interpret Cramer’s V effect size. Which is also reflected in their results by not mentioning it at all. Similarly, there is no reference for eta square for their MANOVA analyses.
Where there any missing data? I would be surprised that there is none, especially when reporting %, one cannot tell, either.
Results
Table 2 has the sample sizes which are redundant since they don’t vary from Table 1.
A chi square analysis to explore a 4x3 analysis, especially when both categories are not ordinal or scalar, is just a free for all. It shows when the authors say “differences in percentage values across the four groups”, to then describe just the highest %. Now, if I follow the subscript letters saying there are “significant differences between groups”, where are those stats? Also, which comparison is being reflected through the table significant testing stats, then? Finally, authors reported 38 significant differences across all 58 items. Yet, the level of describing these results is by saying only how many of comparisons were statistically significant and which group had more or less? No disrespect, but is this the best it can be done?
In addition to the previous comment, are the authors displaying the Benjamini-Hochberg corrected p values in the table? Moreover, why placing the subscript 1 on the “B” letter? Furthermore, why placing a row denoting % when the symbol is in every stat in that table?
On the MANOVA analyses
- A MANOVA is offered, yet there is only indication of the equality of the covariance assumption. All MANOVA assumptions are as follow:
1. Independent Random Sampling
2. Level and Measurement of the Variables
3. Absence of multicollinearity
4. Normality
5. Homogeneity of Variance (addressed)
- authors need to justify why they used Pillai’s Trace, and not other statistical tests
- why do the authors provide the eta squared and not the partial eta squared?
- Table 6 has an “n” for eta, when eta is a similar Greek symbol, yet not the same!
- no noncentrality parameter for each test statistic is displayed. This is a statistical parameter necessary to estimate a noncentral distribution that models the distribution of test statistics if the null hypothesis is not true, and when it is equal to zero, the null hypothesis perfectly fits the data and therefore should not be rejected.
- authors failed to conduct a post-hoc discriminant descriptive analyses in order to determine the mathematical function(s) that distinguish the groups from one another on dependent variable scores. This is pivotal to any MANOVA.
Discussion
Line 262 = Authors can paraphrase the definition of sexual satisfaction. Quotes should be reserved to things that cannot be said if not by direct quotes. A definition is not such case.
Line 266-267 = I thought it could also be both, reward and cost, too?
Line 367 = what kind of “clinical sample” do the author have in mind? Because the sentence has a lot to interpret about it, especially on why it is a construed as a limitation.
Comments on the Quality of English LanguageAuthors are encouraged to perform a thorough review of awkward sentences and adjectives used.
Reviewer 2 Report
Comments and Suggestions for Authors
The authors present a MS on an exchange theory approach.
While I applaud the inclusion of gay/lesbian population, I m not sure in how far the current MS adds to already existing knowledge among straight couples. For gays/lesbians, it is questionable whether the instrument can be used without adaption. Currently, there are items on reproduction and anticonception in the instrument. While they can be applicable to LGBTQI+ populations, the procedure is very different from straight contexts. The authors should reflect on this and potentially adopt their analytical approach.
This manuscript draws on fairly old theorizing, which is in itself not a problem, but it misses out on the last 20 years of research on interpersonal behavior are not reviewed. This is problematic, as the chosen approach is a quite marginal approach. The authors should provide an argument why is the approach is chosen and why it is fitting.
Furthermore, the approach and the instrument used was developed to investigate long-term relationships. The information on the duration of the relationship is missing, and on top of that, is most likely different between straight and gay couples. Also gay couples are often much more dynamic/open/polyamorous, which makes comparisons difficult. The MS should reflect that in a future version.
I am not sure if the current statistical approach is adequate. The current lists include also reproduction/anticonceptions items for LGBTQI+ and can lead to a ranking bias there. Furthermore, I dont think that the endless ranking tables are helpful as an endpoint. The data should be aggregated, for example with a cluster analysis or a random forest, and then plotted. I honestly dont understand the MANOVA presented under 3.3 - what happened to lesbian women? The between gender approach does not make sense for gay men and lesbian women. I would ask the authors kindly to clarify this.
Comments on the Quality of English Languageonly minor editing on run-on sentences is needed
Round 2
Reviewer 1 Report
Comments and Suggestions for Authors
The authors provide a response and corrected every major and minor point addressed by this reviewer. I do however, believe that readers should be be awared that the RCC checklist--for as much as it may be called as a checklist and not questionnaire or scale--is not a psychometric instrument, and that therefore it does not measure a latent construct. Finally, Appendix tables still say % in their column title and give every percentage with a % symbol. But, at this point, it is the authors perrogative.
I leave the manuscript in the hands of the editor to supervise my final comment.
I hereby determine the manuscript to be accepted. Congratulation!
Author Response
Reviewer 1.
The authors provide a response and corrected every major and minor point addressed by this reviewer. I do however, believe that readers should be awared that the RCC checklist-for as much as it may be called as a checklist and not questionnaire or scale--is not a psychometric instrument, and that therefore il does not measure a latent construct. Finally, Appendix tables still say % in their column title and give every percentage with a % symbol. But, at this point, it is the authors perrogative. I leave the manuscript in the hands of the editor to supervise my final comment. I hereby determine the manuscript to be accepted. Congratulation
Response. The authors appreciate your review and suggestions, which have undoubtedly been valuable in improving our manuscript. We agree that the Checklist is not a psychometric instrument to evaluate a latent construct. This information was made explicit in the description of the instrument. We have removed the % symbol in the column title from the Appendix tables. Again, we appreciate your work and the time spent reviewing the manuscript. Thank you.
Reviewer 2 Report
Comments and Suggestions for Authors
I have seen a previous version of the paper and I think that the authors did a good job at addressing my comments. I believe that the MS is stronger now, also with some analyses removed. I still see merit in reducing the complexity of the items and run a cluster or random tree/forest analyses. It would also be helpful to do this for the different letter subgroups in the LGBT. Finally, I believe that the author response letter is argumentatively stronger than some parts of the MS. I would want to invite the authors to improve the MS accordingly and to use this verbiage. The limitation section should be improved along those lines as well.
Author Response
Reviewer 2.
I have seen a previous version of the paper and I think that the authors did a good job at addressing my comments. I believe that the MS is stronger now, also with some analyses removed. I still see merit in reducing the complexity of the items and run a cluster or randam tree/forest analyses. It would also be helpful to do this for the different letter subgroups in the LGBT. Finally, I believe that the author response letter is argumentatively stronger than some parts of the MS. I would want to invite the authors to improve the MS accordingly and to use this verbiage. The limitation section should be improved along those lines as well.
Response. The authors are grateful for the work done, the time invested, and the comments and suggestions to improve our manuscript. Thank you so much. Although performing an analysis of groups or random trees/forests could be useful to summarize our research, we decided to maintain the complete ranking. Conducting such analyses would lead to the loss of information. Furthermore, there is no theoretical justification that we could consider for conducting such analyses. For these reasons, we have not made these changes, nor have we included them as limitations. Without a doubt, doing this study for each population group corresponding to the LGBT subgroups is a great suggestion. This can be a continuation of the present work in order to deepen the study of sexual satisfaction in these populations. Thank you. We hope you understand, and again, thank you very much for your review.